

# Pathogenic strains of *Shewanella putrefaciens* contain plasmids that are absent in the probiotic strain Pdp11

Marta Domínguez-Maqueda[1], Olivia Pérez-Gómez[1],
Ana Grande-Pérez[2,3], Consuelo Esteve[4], Pedro Seoane[5,6],
Silvana T. Tapia-Paniagua[1], Maria Carmen Balebona[1] and
Miguel Angel Moriñigo[1]

[1] Departamento de Microbiología, Universidad de Málaga, Málaga, Spain
[2] Área de Genética, Universidad de Málaga, Málaga, Spain
[3] Instituto de Hortofruticultura Subtropical y Mediterránea "La Mayora"-Universidad de Málaga-Consejo Superior de Investigaciones Científicas (IHSM-UMA-CSIC), Universidad de Málaga, Málaga, Spain
[4] Departmento de Microbiología y Ecología, Universidad de Valencia, Valencia, Spain
[5] Centro de Investigación Biomédica en Red de Enfermedades Raras, CIBERER, Madrid, Spain
[6] Departamento de Biología Molecular y Bioquímica, Universidad de Málaga, Málaga, Spain

Corresponding author
Marta Domínguez-Maqueda,
martadm@uma.es

## ABSTRACT

*Shewanella putrefaciens* Pdp11 is a strain described as a probiotic for use in aquaculture. However, *S. putrefaciens* includes strains reported to be pathogenic or saprophytic to fish. Although the probiotic trait has been related to the presence of a group of genes in its genome, the existence of plasmids that could determine the probiotic or pathogenic character of this bacterium is unknown. In the present work, we searched for plasmids in several strains of *S. putrefaciens* that differ in their pathogenic and probiotic character. Under the different conditions tested, plasmids were only found in two of the five pathogenic strains, but not in the probiotic strain nor in the two saprophytic strains tested. Using a workflow integrating Sanger and Illumina reads, the complete consensus sequences of the plasmids were obtained. Plasmids differed in one ORF and encoded a putative replication initiator protein of the repB family, as well as proteins related to plasmid stability and a toxin-antitoxin system. Phylogenetic analysis showed some similarity to functional repB proteins of other *Shewanella* species. The implication of these plasmids in the probiotic or pathogenic nature of *S. putrefaciens* is discussed.

## INTRODUCTION

Probiotics are live microorganisms that confer a health benefit to the host when administered in adequate amounts (*Sharifuzzaman & Austin, 2017*). Nowadays, their use is increasingly frequent in various sectors such as veterinary, food biotechnology, human health (*Sonnenborn & Schulze, 2009*), or the growing agro-industrial sector (*FAO, 2020*), which includes the aquaculture industry. Probiotics are widely used in aquaculture

species with the aim of increasing the health and productivity of farmed fish as an important source of animal protein (*Assefa & Abunna, 2018*). Several probiotics have been characterized and applied in fish, and some of them are of host origin (*Seegers, Bui & de Vos, 2021*). Such is the case of *Shewanella putrefaciens* Pdp11 strain isolated from skin mucosa of healthy gilthead seabream (*Sparus aurata* L.) (*Chabrillón et al., 2005*; *Díaz-Rosales et al., 2009*) that has been described as a probiotic for farmed fish species such as *Solea senegalensis* and *Sparus aurata* (*Sáenz De Rodrigáñez et al., 2009*; *García de La Banda et al., 2010*; *Tapia-Paniagua et al., 2010*; *Lobo et al., 2014*; *Cordero et al., 2015*). However, *Shewanella putrefaciens* has been associated with diseases in common carp (*Cyprinus carpio* L.) (*Paździor, 2016*), rainbow trout (*Oncorhynchus mykiss*) (*Paździor, Pękala-Safińska & Wasyl, 2019*) and eel (*Anguilla anguilla* L.) (*Esteve, Merchán & Alcaide, 2017*). Therefore, different strains from the same species can be pathogenic or beneficial, as has been shown for *Escherichia coli* (*Sonnenborn & Schulze, 2009*), *Bacillus cereus* (*Cui et al., 2019*) or *Vibrio proteolyticus* (*Bowden, Bricknell & Preziosi, 2018*; *Medina, Moriñigo & Arijo, 2020*), among others. In order to consider a microorganism as a good candidate for probiotic, each specific strain must be thoroughly analysed (*Santos et al., 2020*). A good probiotic must meet a series of essential requirements such as the ability to compete, adhere, persist and survive in the conditions of the intestinal tract (*Ghattargi et al., 2018*). At the same time, probiotics should show absence of virulence factors and multidrug resistance, as undesirable pathogenic traits may appear due to different factors, such as intensive mixed breeding (*Marcos-López et al., 2010*), mutation and recombination *via* horizontal gene transfer (*Wijegoonawardane et al., 2009*), among others. Conventionally, phenotypic, molecular and bioinformatic methods are used to identify genes of interest (virulence/resistance, beneficial…) (*Quainoo et al., 2017*) that may reside either in the bacterial genome or in plasmids or both. Thus, the presence of a group of 15 genes, mostly related to probiotic traits was specifically found in the genome of the probiotic strain *S. putrefaciens* Pdp11, but not in the pathogenic strains SH4, SH6, SH9, SH12 and SH16, nor in the saprophytic strains SdM1 and SdM2 (*Seoane et al., 2019*). Plasmids normally include variable repertoires of 'accessory genes', such as those coding for antibiotic resistance and virulence factors. They also include 'backbone' loci, largely conserved within plasmid families (*Orlek et al., 2017*), such as those involved in key plasmid specific functions (*e.g.*, replication, stable inheritance, mobility) (*Conlan et al., 2014*; *Giess et al., 2016*). Therefore, plasmid characterization could help to discern whether a microorganism is pathogenic or beneficial (*Santos et al., 2020*).

In the present work, we searched for plasmids in several strains of *S. putrefaciens* that differed in their pathogenic or probiotic character. Plasmids were only found in two of the five pathogenic strains and not in the probiotic or the two saprophytic strains tested. Using a workflow integrating Sanger and Illumina sequences, the complete consensus sequences of these plasmids were obtained. Sequence analysis showed that the plasmids encoded a putative replication initiator protein of the repB family, and proteins related to plasmid stability and to a toxin-antitoxin system. Phylogenetic analysis showed similarity with other *Shewanella* species in functional repB proteins. The implication of these plasmids in the probiotic or pathogenic nature of *S. putrefaciens* is discussed.

## MATERIALS AND METHODS

### Bacterial strains and growth conditions

*Shewanella putrefaciens* strains SH4, SH6, SH9, SH12 and SH16 isolated from diseased eels were kindly provided by Dra. Esteve C. from the University of Valencia (Valencia, Spain) (*Esteve, Merchán & Alcaide, 2017*). Two saprophytic isolates (SdM1 and SdM2) of *S. putrefaciens* were obtained from environmental sources (*Seoane et al., 2019*) and one probiotic strain, *S. putrefaciens* Pdp11 CECT 7627, was isolated from the skin of healthy *Sparus aurata* L. (*Díaz-Rosales et al. 2009*).

All *S. putrefaciens* strains were grown on tryptic soy agar plates (Oxoid Ltd., Basingstoke, UK) supplemented with NaCl (1.5%) (TSAs) for 24 h at 23 °C. Then, one or two colonies of each strain were picked and grown in 10 mL tubes of tryptic soy broth (Oxoid Ltd., Basingstoke, UK) added with NaCl (1.5%) (TSBs) for 24 h at 23 °C on shaking at 80 rpm (ELMI DOS-20M Digital Orbital Shaker, USA). Given the possibility that, if present, plasmids could be integrated into the bacterial chromosome, strains were cultured under different growth conditions (temperature, incubation time, growth medium and freeze-thaw) to favour their excision. For this, one or two colonies of each strain were picked and cultured in 10 mL tubes of TSBs and minimal (M9) media, and incubated at 23 °C or 4 °C for 24 and 48 h on shaking at 80 rpm. Cultures inoculated in parallel with TSBs medium containing glycerol (20%) were subjected to a freeze-thaw cycle (*Pesaro et al., 2003*) for 24 h at −80 °C prior to incubation.

### Plasmid DNA isolation

One or two colonies of the *S. putrefaciens* strains grown under different growth conditions were picked from the pure culture and grown in 10 mL of TSBs (Oxoid Ltd., Basingstoke, UK) for 24 h at 23 °C under agitaton at 80 rpm (ELMI DOS-20M Digital Orbital Shaker). As a positive control, *Escherichia coli* V157, a strain harboring seven plasmids (*Macrina et al., 1978*), was grown on Luria-Bertani agar plates (LB) (Becton, Dickson and Company, Le Pont de Claix, France) for 24 h at 37 °C. The cultures were centrifuged at 8,000×*g* for 5 min and the pellet was used for plasmid DNA (pDNA) isolation using the GeneJet Plasmid Miniprep kit (Thermo Fisher, Waltham, MA, USA) following the manufacturer's protocol. Plasmid DNA integrity was checked by agarose gel (0.8%, w/v) electrophoresis in the presence of RedSafe nucleic acid staining solution (InTRON Biotechnology, Seongnam-Si, South Korea). The 0.2–10 kb Hyperladder molecular weight marker (Bioline, Taunton, MA, USA) was used to check plasmid size. The pDNA was stored at −20 °C for further processing.

### Plasmid DNA amplification and sequencing

Rolling circle amplification (RCA) was performed using the TempliPhi 100 amplification kit (GE Healthcare, Chicago, IL, USA) for each isolated plasmid following the manufacturer's instructions. As a positive control for the reaction, pUC19 was used. DNA from each RCA-amplified plasmid was digested with EcoRI and EcoRV (Takara, San Jose, CA, USA), separately. They were then ligated employing T4 DNA ligase (Thermo Fisher Scientific, Waltham, MA, USA) to pBluescript SK (+) (pBSK S+) (Addgene,

London, UK) previously treated with shrimp phosphatase alkaline (New England Biolabs, Ipswich, MA, USA). Plasmid DNA was transformed into $CaCl_2$-treated *E. coli* DH5α competent cells, as described by *Sambrook, Fritsch & Maniatis (1989)*. Recombinant bacteria were selected using plates containing ampicillin (100 μg/ml) and X-gal (20 μg/ml), and colonies resuspended in 50 μl of DEPC-Water (Sigma, St. Louis, MI, USA) and tested by PCR amplification. Reaction mixture contained 2 μl of bacterial suspension, 62,5 U of Taq Accustart II Trough Mix (Boimerieux, Marcy-l'Étoile, France), 20 pmol of M13R primer 5′-CAGGAAACAGCTATGAC-3′ and 20 pmol of M13F primer 5′-TGTAAAACGACGGCCAGT-3′ in a final volume of 10 μL. The PCR program was: (1) 94 °C 3 min, (2) 94 °C 30 s, (3) 50 °C 30 s, (4) 72 °C 1 min/kb. Steps (2) to (4) were repeated during 35 cycles. Amplicons were checked by agarose electrophoresis gel as above. PCR products were sent to Macrogen (Madrid, Spain) for Sanger sequencing.

## Plasmid DNA sequence assembly

Plasmids were assembled *de novo* using a workflow integrating Sanger and Illumina reads (*Chevreux, 2005*; *Bankevich et al., 2012*). Plasmid-specific Illumina sequences used here were obtained by *Seoane et al. (2019)* using a sequencing library with the Nextera protocol for 2 × 300 bp PE (raw reads available in BioProject PRJNA510237).

First, the Sanger plasmid sequences were pre-processed with BBDuk (*Bushnell, 2015*; *Kechin et al., 2017*) a tool developed for quality filtering and adapter trimming using k-mers matching, where k-mers are substrings of length "k" contained in the original nucleotide sequence. To remove the sequence of the pBSK S+ vector, different values of the two clean-up parameters k and hdist were applied, where "k" is the length of the k-mers and "hdist" is the "hamming distance" between two k-mers. We used a k = 20 and hdist = 1 for pSH12 and a k = 22 and hdist = 2 for pSH4 (Fig. 1A). Two sets of vector-free Sanger reads were generated when quality trimming with BBDuk was applied using "trimq" parameter. This parameter is the minimum average quality required in a sequence window, if the average is below the threshold, the nucleotides are trimmed from the read. For each plasmid, a low (10) and a high (17) stringency quality values were applied. The low stringency threshold generate longer reads that are used to capture Illumina reads from the BioProject PRJNA510237, denoted as Capture read set (Fig. 1B). With the high strigency threshold, a high quality read set for the final assembly is generated, denoted as Assembly read set (Fig. 1C).

Next, Illumina plasmid reads were captured with the sanger Capture read set using bowtie 2 (*Langmead & Salzberg, 2012*; *Langmead, 2017*). These reads were identified and subsequently extracted with Samtools (*Li et al., 2009*) by selecting pairs of Illumina reads in which at least one member had aligned to the Capture read set (Fig. 1D). Then, Sanger-Illumina hybrid assembly was performed with two assemblers, Mira (*Chevreux, 2005*) and Spades (*Bankevich et al., 2012*) using default parameter values. The aforementioned Illumina reads and the Assembly read set, obtaining two possible and complete pDNA assemblies per strain (Fig. 1E). The two sequences per strain were recircularized by the MARS method (*Ayad & Pissis, 2017*) (Fig. 1F) to make possible the next step. The plasmid sequences of each strain, they were aligned by the Clustal method

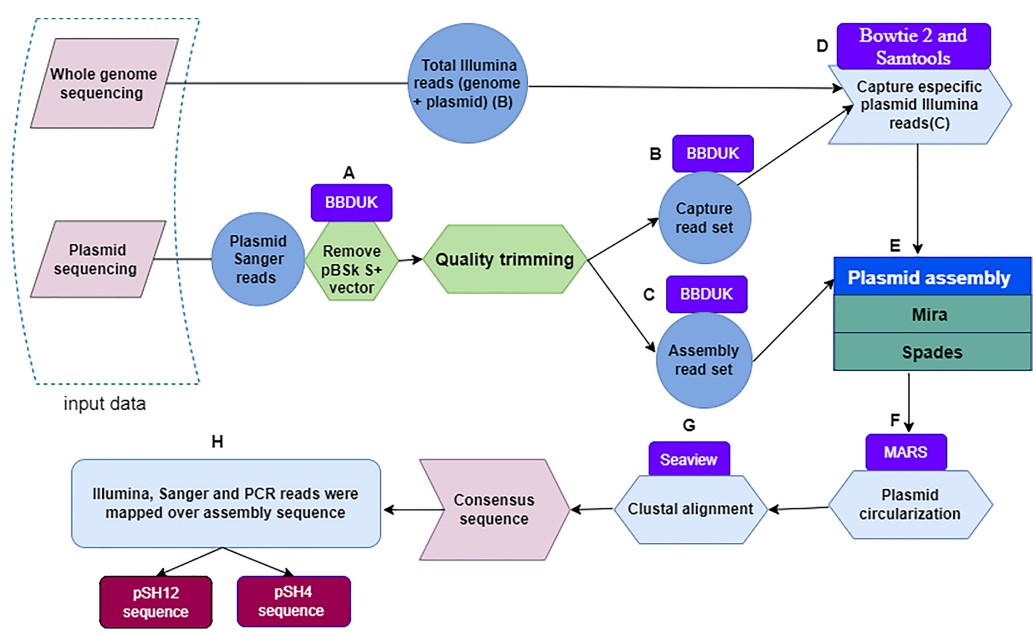

**Figure 1 Flowchart depicting the workflow used to obtain fully assembled plasmid DNA sequences from *S. putrefaciens* strains SH4 and SH12.** Within a dashed box, the two sources of plasmid sequences are shown. Quality trimming steps (A) performed with the BBDuk software leading to capture read set are indicated by the letters (B), and those leading to assembly read set by the letters (C). The Bowtie and Samtools programs, denoted with the letter (D), were used to capture specific Illumina sequences. The output of the Mira and Spades programs was a sequence generated from Illumina and Sanger reads (E), and circularized by the MARS method (F). Plasmids were aligned by the Clustal method of the Seaview program (G) to finally obtain the plasmid consensus sequences. The consensus sequence was mapping by sanger, illumina and PCR reads for assembly validation.

(*Sievers & Higgins, 2014*) of the Seaview program (*Galtier, Gouy & Gautier, 1996*) (Fig. 1G), leading to a unique plasmid consensus sequence per strain.

## Assembly validation

Alignment of the sequence set of SH4 and SH12 plasmids using Mira and Spades programs revealed the presence of unreliable regions in both plasmids. Validation was carried out by PCR and primers were designed for each non-validated plasmid sequence using Primer3 (*Untergasser et al., 2012*). The composition of the reaction mixture was done as above, except that 10 pmol of the primers pSH4_R 5′<CGGATTGAATGGCTGGCTGGACTG>3 (reverse) and pSH4_F 5′<ATACCAAACGCCCACAGA>3′ (direct) were used to validate the assembly of pSH4, and of the primers pSH12_R 5′<GGCTCCACCCTTACCCAAAAAA>3′ (reverse) and pSH12_F <5′GCGAGCCCCTCCATGATTTT>3′ (direct) to validate the assembly of pSH12. The PCR program was: (1) 94 °C 3 min, (2) 94 °C 30 s, (3) 66 °C 30 s, (4) 72 °C 1 min/kb, with 35 repeat cycles of steps (2) through (4). PCR products were checked by agarose gel electrophoresis using the 100 bp molecular weight marker Hyperladder (Bioline, Taunton, MA, USA). Finally, the plasmid assemblies of strains SH4 and SH12 were mapped back to the Sanger assembly read set sequence, the Illumina

sequences captured from Sanger capture read set sequence and the PCR product obtained from validation.

## Plasmid annotation and functional characterization

Once complete consensus sequences were obtained for each plasmid, sequences were searched for on the PLSDB web server (*Galata et al., 2019*), a resource containing numerous plasmid records collected from the NCBI nucleotide database. Next, gene prediction was performed for annotation of bacterial operons and open reading frames (ORFs) using Softberry Software (http://www.softberry.com/berry.phtml) including the fgenesB tool (*Solovyev et al., 2011*) with "Bacterial generic" as the closest organism; the Prokaryotic GeneMark. Hmm version 2 program (*Besemer, Lomsadze & Borodovsky, 2001*) with "*Shewanella putrefaciens* CN_32" as the selected species. Consensus sequences were also analysed with ORF finder at NCBI website (https://www.ncbi.nlm.nih.gov/orffinder/) with the search parameters: 150 nucleotides of minimal ORF length, genetic code "Bacterial, Archaeal and Plant Plastid", and "ATG and alternative initiation codons" as ORF start codon to be used. For each of the identified ORFs, their amino acid sequence was obtained and queried in blastp (NCBI, Bethesda, Maryland, USA), to obtain clues about the ORF functions (*Altschul et al., 1990*; *Gish & States, 1993*). PHYRE 2 version 2.0, was also used for subsequent protein prediction and modelling (*Kelley et al., 2015*). Finally, Dfast (*Tanizawa, Fujisawa & Nakamura, 2018*) was used for ORF protein identification and functional annotation. Plasmid maps were drawn with pDRAW32 version 1.1.146 (AcaClone software, Greeley, CO, USA) (https://www.acaclone.com/).

## Plasmid to genome comparison

To detect the degree of similarity between the genomes of the SH4 and SH12 strains and their respective plasmids, a comparison between them was carried out using Sibelia (*Minkin et al., 2013*). The complete putative pDNA sequences and their corresponding ORFs were also searched by blastn against the probiotic genome and all non-probiotic strains (NPS) genomes previously described by *Seoane et al. (2019)*.

## Phylogenetic study

The repB protein sequences of pSH12 and pSH4 were searched in Blast (NCBI, Bethesda, Maryland, USA). The repB protein sequences of plasmids from different bacterial species were downloaded in FASTA format in a single export file, and alignment was performed by the Clustal method (*Sievers & Higgins, 2014*). A phylogenetic tree was constructed by PhyML employing the Maximum Likelihood method (*Edwards & Cavalli-Sforza, 1964*) using Seaview software (*Galtier, Gouy & Gautier, 1996*).

## Ethical statement

To conduct the research, we used bacterial strain *Shewanella putrefaciens* (strains SH4, SH6, SH9, SH12, SH16, SdM1, SdM2 and Pdp11), which is not considered a human or animal sample. Therefore, no ethics approval was needed, and no informed consent was required.

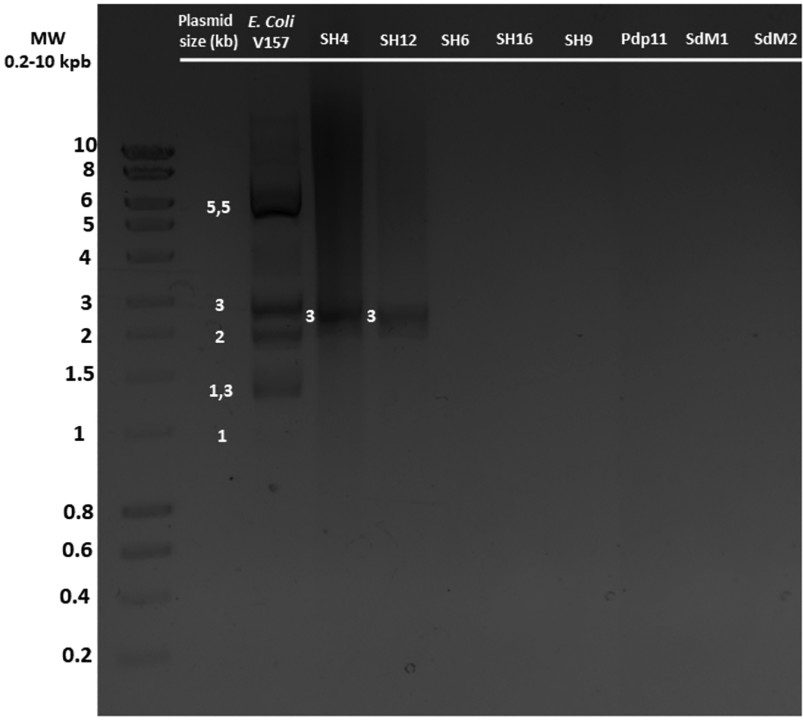

**Figure 2 Plasmid DNA isolated from different *S. putrefaciens* strains (SH4, SH12, SH6, SH16, SH9, Pdp11, SdM1 and SdM2) separated by agarose gel electrophoresis.** All strains were grown under the optimal growth conditions of 23 °C in TSBs under shaking for 24 h. The sizes of the individual plasmids contained in strains SH4 and SH12 as well as the five plasmids detected of the positive control *E. coli* O157, grown in LB broth, are indicated.

## RESULTS

### Identification of plasmids

*S. putrefaciens* strains SH4, SH6, SH12, SH16, SH19, Pdp11, SdM1 and SdM2 were analysed to confirm the presence or absence of plasmids under different growth conditions (Table S1). After growth under optimal conditions, only SH4 and SH12 strains were found to contain one plasmid around 3 kb in size (Fig. 2). The same results were obtained when other growth conditions were used (Table S1). *E. coli* V157 strain, used as a positive control, showed all seven plasmids previously described in the literature (*Macrina et al., 1978*).

### Plasmid assembly and sequence validation

The consensus sequences of pSH4 and pSH12 obtained using Mira-Spade workflow from Sanger and Illumina reads showed a size of 3,003 bp for pSH4 and 2,990 bp for pSH12. However, unvalidated regions were detected in both plasmids (Fig. 3) because the Capture read set sequence did not sufficiently capture the Illumina sequences in pSH4 (Fig. 3A) and pSH12 (Fig. 3B), rendering these regions unreliable. Moreover, the Assembly read set sequence did not support this region in the alignment of pSH4 (Fig. 3A), unlike in pSH12 (Fig. 3B). From the PCR product obtained, a new alignment using Sanger, Illumina and PCR product sequences was carried out for each plasmid and mapped. After this, the

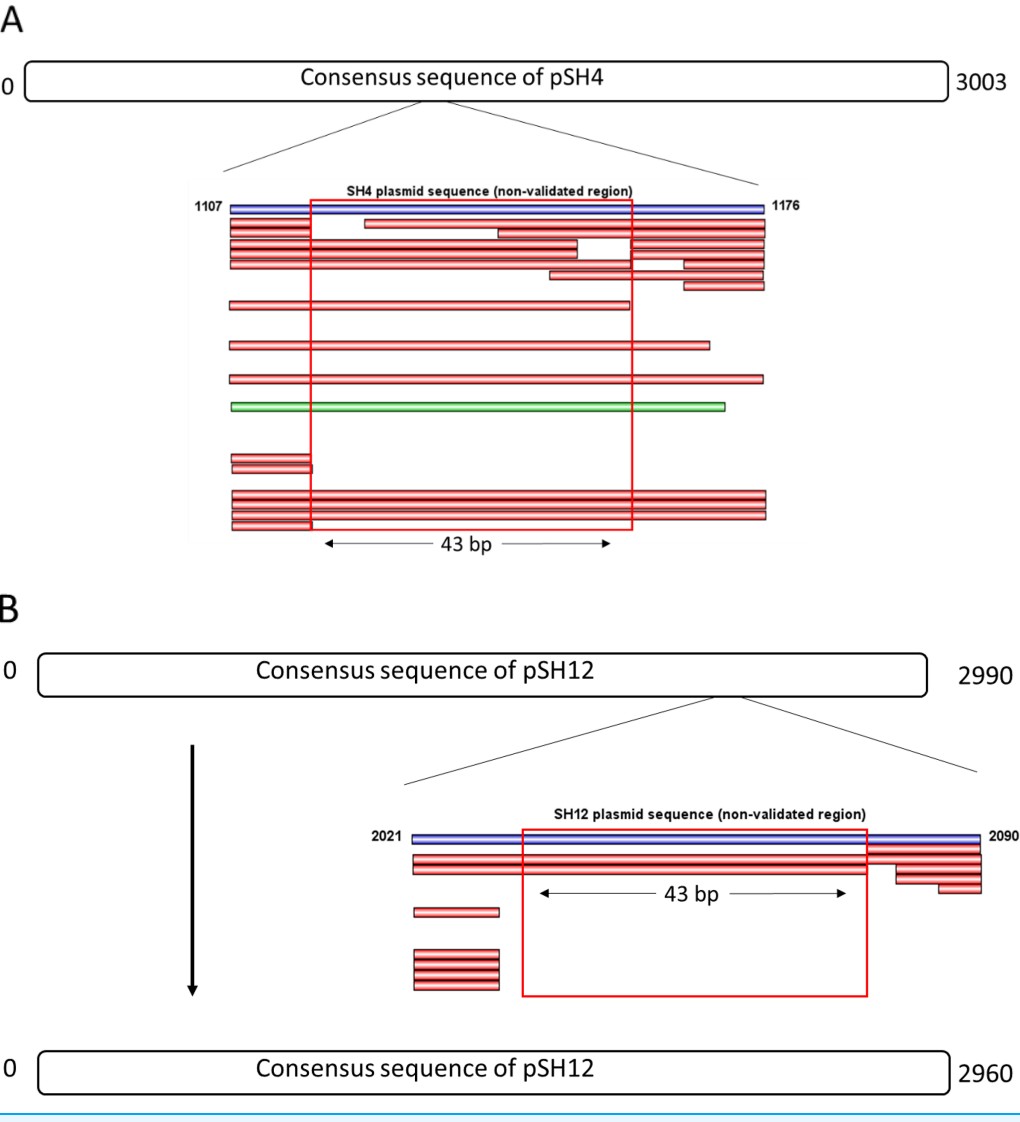

**Figure 3 Complete consensus sequences of pSH4 (A) and pSH12 (B) showing the unvalidated regions (blue line) resolved by mapping with Illumina and Sanger sequences.** Both unvalidated regions in the consensus sequences were subjected to validation by PCR amplification (A) The unvalidated region of pSH4 supported by 22 Illumina sequence (red bars) and one Sanger sequence (green bar) led to a reliable sequence between positions 1,107 and 1,176 in the pSH4 consensus sequence. (B) The unvalidated region of pSH12, located between nucleotides 2021 and 2090, was poorly supported by Illumina sequences and no Sanger sequences, so the consensus sequence of pSH12 turned out to be 30 nucleotides shorter than that of SH4.

unvalidated region of pSH4 was supported by a total of 22 Illumina and one Sanger sequences as was the PCR amplicon, while that of pSH12 was invalid. Thus, fully assembled circular sequences showed 3,003 bp in pSH4 but 2,960 bp in pSH12, 43 pb less due to the absence of that unreliable region (Fig. 3B). The GC content of both plasmids was similar, 42.69% (pSH4) and 43.04% (pSH12). The raw sequence information from Seaview program and agarose gel of PCR amplicons are showed in Fig. S1.

**Table 1 Common ORFs found in pSH4 and pSH12 in all predictions tools tested.**

| Strain | ORF[a] | Strand | Start | Stop | Lenght (nt \| aa) | G+C content (%) |
|--------|------|--------|-------|------|------------------|-----------------|
| SH4 | 3 | + | 2,109 | 2,420 | 312 \| 103 | 36.74 |
| | 5 | – | 1,541 | 1,158 | 384 \| 127 | 57.22 |
| | 6 | – | 1,288 | 728 | 561 \| 186 | 44.15 |
| | 7 | – | 727 | 275 | 453 \| 150 | 44.35 |
| SH12 | 2 | + | 2,143 | 2,454 | 312 \| 103 | 42.47 |
| | 5 | – | 1,575 | 889 | 687 \| 228 | 46.44 |
| | 6 | – | 888 | 436 | 453 \| 150 | 48.76 |

Note:
[a] ORFs were named as ORFn followed by the plasmid name (pSH12 or pSH4), where n is the ORF id.

## ORF analysis of pSH4 and pSH12

No hits sequences of the plasmids were available on the PLSDB web server (*Galata et al., 2019*), so a comprehensive ORF analysis was carried out. Analysis using the ORF finder revealed 10 possible ORFs in pSH4 and nine in pSH12. However, after removing nested predictions (*i.e.*, overlapping ORFs) a total of seven ORFs remained in pSH4 and six in pSH12 (Table S2). The gene prediction tools, fgenesB, GeneMark and ORF finder, confirmed that only four and three ORFs in pSH4 and pSH12, respectively, were common and highly similar in both plasmids (Table 1).

Both plasmid sequences resulted in the presence of the same ORFs, except for the presence of an extra 43 bp region in pSH4 that introduced a stop codon that split in two the equivalent of ORF5 in pSH12 giving rise to ORF5 and ORF6 in pSH4. The localization and orientation of all ORFs of pSH12 and pSH4 are shown in the circular map in Fig. 4. Since unique EcoRI and EcoRV cleavage sites were found in both plasmids, they were used for cloning and sequencing.

## Prediction of functional proteins

Based on blastp and Phyre 2, ORF5 and ORF6 of pSH4, and ORF5 of pSH12 presented a high identity to a plasmid replication initiator protein of the repB family of *Shewanella algae* (Table 2). The proteins encoded by these ORFs were predicted to show structural analogy with a crystal structure of the plasmid initiator protein of the repB family, with a nucleic acid as ligand and functions such as DNA replication and plasmid copy control (Table 2). Similarly, the proteins encoded by ORF7 of pSH4 and ORF6 of pSH12 showed analogy with the Type II toxin-antitoxin system PemK/MazF family toxin from *Avibacterium paragallinarum*, presenting DNA as ligand and functions of cell growth inhibitor by cleaving mRNA with high sequence-specificity as part of the bacterial stress response, and as toxic component for plasmid maintenance (Table 2). No reliable predictions were obtained for the remaining ORFs that showed similarities to hypothetical or unknown proteins.
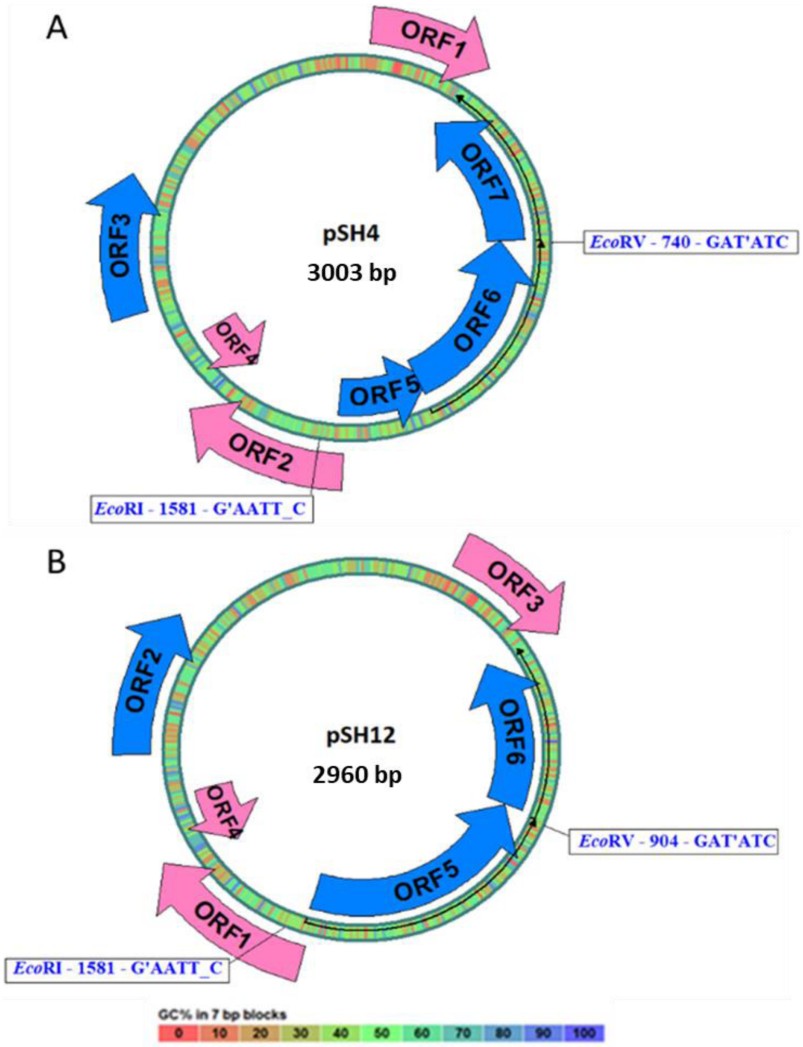

**Figure 4  Circular map of pSH4 (A) and pSH12 (B).** The main circle that shapes the plasmid is coloured according to the percentage of G+C shown in the bottom bar. Solid arrows indicate the approximate position of the ORF and the orientation of the transcript. ORFs encoded by the (+) strand of the nucleotide sequence are outside the circle and ORFs encoded by the (−) strand inside. The range of ORF colours on the map reveals differences according to the analysis of the putative translation products and their respective functions. The blue colour corresponds to informative ORFs, whose amino acid sequences showed similar homologies with partial proteins of known function and hypothetical proteins in both plasmids, according to protein searches. In addition, they are labeled as complete genes by the algorithms used. The rest of ORFs are drawn with pink arrows.

## Genomic and plasmid comparison between strains

Comparison of the genomes of SH4 and SH12 and their respective plasmids yielded high similarity at the plasmid level with 94.24% coverage (Fig. S2), differing only by an additional 43 pb region in pSH4 as explained above. At the same time, genome-wide comparison of SH4 and SH12 strains also showed high similarity composed of 46 and 58 scaffolds, respectively with 99.42% coverage between each other (Fig. S2). These

**Table 2 Functional annotation of the main ORFs of pSH4 and pSH12 by blastp and Phyre2.**

| Strain | Blastp | | | | | | Phyre 2 | |
| --- | --- | --- | --- | --- | --- | --- | --- | --- |
| | ORF | Accesion ID | Protein | Identity (%) | Coverage (%) | Organism | Superfamily/family | Confidence |
| SH4 | 3 | MBP7664129.1 | Hypothetical protein | 78.48 | 94.00 | *Shewanella* sp | Family: BH3980-like | 29.9 |
| | 5 | WP_208142329.1 | repB family plasmid replication initiator protein | 76.92 | 89.00 | *Shewanella algae* | Family: Replication initiation protein | 100 |
| | 6 | WP_208142329.1 | repB family plasmid replication initiator protein | 67.74 | 100.00 | *Shewanella algae* | Superfamily: "Winged helix" DNA-binding domain Family: Replication initiation protein | 100 |
| | 7 | WP_115615842.1 | Type II toxin-antitoxin system PemK/MazF family toxin | 54.17 | 96.00 | *Avibacterium paragallinarum* | Superfamily: Cell growth inhibitor/plasmid maintenance toxic component Family: Kid/PemK | 99.4 |
| SH12 | 2 | NCS64480.1 | Hypothetical protein | 87.18 | 92.00 | *Thiomicrospira* sp. | BH3980-like | 29.9 |
| | 5 | WP_208142329.1 | repB family plasmid replication initiator protein | 72.89 | 100 | *Shewanella algae* | Superfamily: "Winged helix" DNA-binding domain Family: Replication initiation protein | 100 |
| | 6 | WP_115615842.1 | Type II toxin-antitoxin system PemK/MazF family toxin | 54.17 | 96.00 | *Avibacterium paragallinarum* | Superfamily: Cell growth inhibitor/plasmid maintenance toxic component Family: Kid/PemK | 99.4 |

percentage values support the idea that SH4 and SH12 are different strains belonging to the same species.

In addition, a blastn alignment was carried out to check for a possible insertion of these plasmids or any of their ORFs in the complete genome of the other strains under study. A total absence of plasmids was found in the rest of NPS genomes, as wells as in the genome of the probiotic *S. putrefaciens* Pdp11.

## Phylogenetic analysis of the repB protein

The repB protein sequences found in pSH12 (ORF5) and pSH4 (ORF5 and ORF6) were compared extensively and a phylogenetic analysis was performed (Fig. 5) to look for similar disruptions in the repB sequences and their possible effects on functionality. The disruption in the repB protein in pSH4 gave rise to ORF5 and ORF6, two short repB sequences encoding 127 and 186 amino acids, respectively, indicating that the ORF6 was the longest conserved region of the repB protein in pSH4, close to the only 228 amino acid protein encoded by ORF5 in pSH12. Phylogenetic analysis showed that the pSH12 protein was conserved and structurally similar to other repB proteins described in *Shewanella* species (Fig. 5). In addition, the repB proteins of both plasmids also were closely related to the repB protein of *Pseudoalteromonas* sp, *Thiomicrospira* sp. and other *Shewanella* species (Fig. 5).

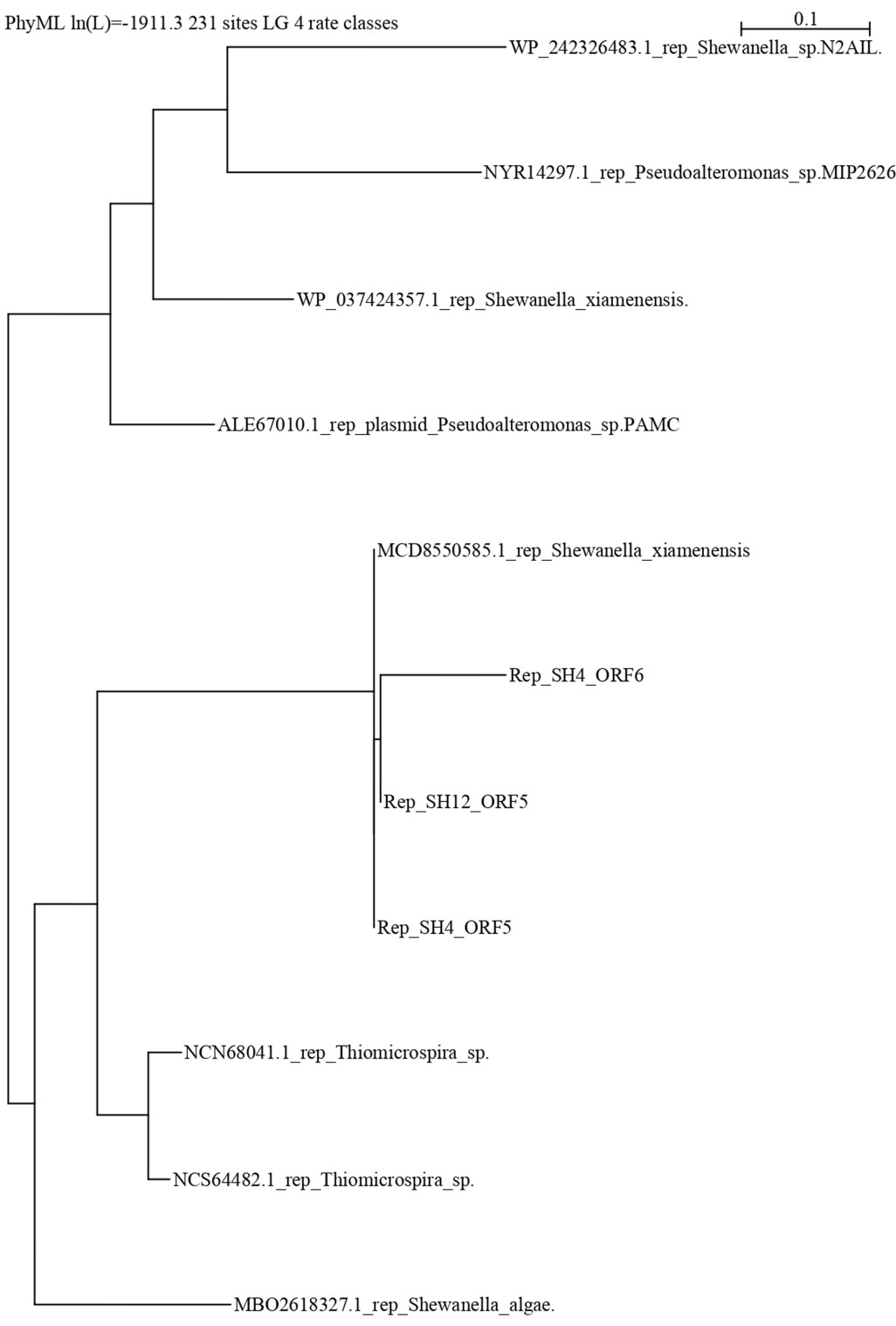

**Figure 5 Phylogenetic tree using repB protein sequences of pSH12 and pSH4.** The different protein sequences were downloaded from the NCBI database, and alignment was performed by Clustal method followed by PhyML algorithm to infer distances by using the Seaview software.

## DISCUSSION

The reason why different strains of the same bacterial species can be either pathogenic or probiotic remains elusive, but knowledge of the molecular basis may help to understand it (*Seoane et al., 2019*). In this work, the study of plasmids from different strains of *Shewanella putrefaciens*, a bacterium of commercial and biotechnological importance for aquaculture, has been addressed.

The presence of plasmids in the *Shewanella* group is often associated with resistance traits, as is the case for plasmid pSFKW33 of *Shewanella* sp. 33B (*Werbowy, Cieśliński & Kur, 2009*), plasmid pSMX33 of *Shewanella xiamenensis* BC01 (*Zhou & Ng, 2016*) or plasmid pSheB, of *Shewanella* sp. O23S (*Uhrynowski, Radlinska & Drewniak, 2019*). The horizontal mobile genetic element (MGE) products encoded by plasmids could be at the root of the differences between probiotic and pathogenic strains, as has been described for the *Enterococcus* group (*Santos et al., 2020*). Therefore, the absence of plasmids in *S. putrefaciens* strain Pdp11 could underlie its probiotic nature. However, this fact does not guarantee probiosis, since three of the five pathogenic strains tested (strains SH6, SH9 and SH16) did not contain any plasmids. Given that plasmids were only detected in two of the five pathogenic strains, it cannot be ruled out that virulence factors in *S. putrefaciens* are chromosomally encoded nor that no other plasmids were detected because of the extraction methodology used in this work. The pathogenic strains SH4 and SH12 had one plasmid each, pSH4 and pSH12, respectively, which did not show much similarity to the plasmids in the databases. Only two ORFs with a high percentage of identity to known plasmid proteins were identified, while the rest of the ORFs belonged to hypothetical or unknown proteins. Since only 1% to 29% of plasmid sequences are usually found in bacterial databases (*Maguire et al., 2020*), especially in the genus *Shewanella*, their characterization is of interest. In this work, we found that the main ORFs conserved in both plasmids pSH4 and pSH12 belong to the replication protein initiator repB superfamily, and to the type II toxin-antitoxin system superfamily (PemK/PemI family protein).

Rep proteins are especially important as they are primarily responsible for the initial DNA binding and nicking activities, which represent the first steps in plasmid replication and conjugative plasmid transfer (*Stolz, 2014*). It is generally composed of repA (helicase), repB (initiator protein) and repC (initiator protein) proteins (*Wawrzyniak et al., 2019*) but only repB protein has been identified in pSH4 and pSH12. Some bacterial strains such as *Agrobacterium tumefaciens* require the products of the repABC operon for plasmid replication, as they encode its rep proteins (*Chai & Winans, 2005*; *Pinto, Pappas & Winans, 2012*). Since these proteins recruit several other DNA polymerases and helicases from the bacterial host to proceed with plasmid replication (*San Millan & Craig Maclean, 2019*), it seems plausible that SH4 and SH12 may use this mechanism to replicate. In addition, repB protein in pSH4 supposed major attention due to the disruption of it, which can affect its oligomerization and functionality as occurs in rep proteins Rep68/78 as described (*Zarate-Perez et al., 2012*). The results of the phylogenetic analysis suggest that

the larger and more conserved ORF5 may retain its functionality, whereas future studies will be necessary to determine the functionality of ORF6.

In any case, the identification of the repB protein is interesting because it has been previously explored in *S. xiamenensis* BC01 and *S. oneidensis* MR-1 (*Zhou & Ng, 2016*) as functional ori for stable plasmid replication in *Shewanella*. Furthermore, the combination of repB with other promoters such as placI favoured the expression of *gfp* and *mtr* genes by recombinant DNA technology in *S. oneidiensis* MR-1 (*Ng et al., 2018*). In this way, specific plasmids could be constructed and incorporated into other *Shewanella* species to favour the expression of exogenous genes of interest (*Antonio-Hernández et al., 2019*). According to this compatibility, the above application could be attractive for future studies to improve the toolbox of the probiotic strain Pdp11.

Another major ORF identified is the PemK protein, which is part of the type II PemK-PemI toxin-antitoxin (TA) system. This system is found both in bacterial chromosomes and in MGEs such as plasmids and prophages (*Bukowski et al., 2019*). A type II TA system typically consists of two genes located in an operon encoding a stable toxin that disrupts essential cellular processes and a labile antitoxin that forms a tight protein complex with the cognate toxin to neutralize its activity (*Yao et al., 2018*). Activities that have been associated with type II TA systems are maintenance of genetic material, bacterial virulence, biofilm formation, phage inhibition, and different types of stress management, including antibiotic tolerance and persister formation (*Bleriot et al., 2020*). In this work, the gene encoding the PemK protein has been detected in the plasmids of pathogenic strains of *Shewanella*, suggesting that its unstable antitoxin could consist of a protein or a non-coding RNA (*Yamaguchi, Park & Inouye, 2011*). On the other hand, strains SH4 and SH12 might not carry the complementary antitoxin PemI, allowing the toxin to exert its virulent effect. Furthermore, since virulence and drug-resistant genes had been detected in different species of *Shewanella*, such as *S. algae*, *S. putrefaciens*, *S. xiamenensis*, *S. oneidensis* and *S. frigidimarina*, mainly on the chromosome associated or not with MGEs (*Yousfi et al., 2017*), a genomic and plasmid comparison of SH4 and SH12 with the rest of the study strains was performed to check the presence of these plasmid sequences and in particular, of the TA system. The results allowed to exclude the probiotic Pdp11 since it lacks the TA type II system as a virulence factor and its self-regulatory characteristics.

Phenotypic and pathogenic differences had been identified between the pathogenic strains SH4 and SH12. The lethal doses (LD50) of strains SH4 and SH12 were $2.4 \times 10^6$ and $2.8 \times 10^6$, respectively, in addition to obvious differences in metabolic characteristics between these two strains (*Esteve, Merchán & Alcaide, 2017*). However, taking into account previous results on phenotypic (*Esteve, Merchán & Alcaide, 2017*) and genomic (*Seoane et al., 2019*) characteristics, our results seem to indicate a greater similarity between both strains, and may even suggest considering them as possible clones with specific differences. In any case, the absence in Pdp11 of plasmids carrying virulence genes supports the idea of its probiotic nature and opens the way for future research on biotechnological applications. To comply with EU regulations on GMOs, these could be aimed at obtaining a higher probiotic value through natural processes (conjugation,

transformation and transduction). The use of genetic engineering to improve derived products such as metabolites and extracellular products, among others, could also be considered.

## CONCLUSIONS

The presence of plasmids is expected to be associated with bacterial survival and, commonly, with virulence factors. The present work has characterized two new plasmids present in two strains of *S. putrefaciens*, SH4 and SH12, which despite having similar genomic and plasmid profiles with high levels of identity, turned out to be different strains of the same species. The plasmids identified encode several proteins of known or unknown function. Among the former, two were found, one belonging to an essential protein, repB, necessary for plasmid replication, and the other, pemK, corresponding to a virulence factor. The probiotic strain *S. putrefaciens* Pdp11 did not present plasmids, which may be behind its probiotic nature, making it unique in comparison to other *Shewanella putrefaciens* strains.

## ACKNOWLEDGEMENTS

The authors are grateful for the computing resources and technical support provided by the Plataforma Andaluza de Bioinformática of the University of Malaga.

### Funding

This study has been supported by MINECO and co-financed with FEDER funds (Grant AG-2017-509 83370-C3-3-R). Ana Grande-Pérez received a grant awarded to the BIO-264 Research Group by the Consejería de Economía, Innovación y Ciencia, Junta de Andalucía, with the support of the European Regional Development Fund (ERDF) and the European Social Fund (ESF). The funders had no role in study design, data collection and analysis, decision to publish, or preparation of the manuscript.

### Grant Disclosures

The following grant information was disclosed by the authors:
MINECO.
FEDER: AG-2017-509 83370-C3-3-R.
European Regional Development Fund (ERDF).
European Social Fund (ESF).

### Competing Interests

Ana Grande Pérez is an Academic Editor for PeerJ.

### Author Contributions

- Marta Domínguez Maqueda performed the experiments, analyzed the data, prepared figures and/or tables, authored or reviewed drafts of the article, and approved the final draft.

- Olivia Pérez Gómez performed the experiments, analyzed the data, prepared figures and/or tables, authored or reviewed drafts of the article, and approved the final draft.
- Ana Grande-Pérez conceived and designed the experiments, performed the experiments, analyzed the data, authored or reviewed drafts of the article, and approved the final draft.
- Consuelo Esteve conceived and designed the experiments, authored or reviewed drafts of the article, provided the pathogenic strains, and approved the final draft.
- Pedro Seoane performed the experiments, analyzed the data, authored or reviewed drafts of the article, and approved the final draft.
- Silvana T. Tapia-Paniagua conceived and designed the experiments, performed the experiments, authored or reviewed drafts of the article, and approved the final draft.
- Maria Carmen Balebona conceived and designed the experiments, authored or reviewed drafts of the article, and approved the final draft.
- Miguel Angel Moriñigo conceived and designed the experiments, authored or reviewed drafts of the article, and approved the final draft.

### DNA Deposition

The following information was supplied regarding the deposition of DNA sequences:

The sequencing data is available at NCBI: PRJNA798051; SAMN25041459 (pSH4 sequence) and SAMN25041460 (pSH12 sequence).

### Data Availability

The data is available at Zenodo (oliviaastencio. (2022). oliviaastencio/shewanella_plasmid: v1 (v1.0). Zenodo. https://doi.org/10.5281/zenodo.7092606) and GitHub: https://github.com/oliviaastencio/shewanella_plasmid).

### Supplemental Information

Supplemental information for this article can be found online at http://dx.doi.org/10.7717/peerj.14248#supplemental-information.

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
