# Peer review of "Pathogenic strains of Shewanella putrefaciens contain plasmids that are absent in the probiotic strain Pdp11"

_PeerJ, doi:10.7717/peerj.14248_

## Round 0.1 · original submission · Major Revisions

Dear Dr. Maqueda and colleagues:

Thanks for submitting your manuscript to PeerJ. I have now received two independent reviews of your work, and as you will see, the reviewers raised some minor concerns about the research (mostly the manuscript format and content). Despite this, these reviewers are optimistic about your work and the potential impact it will have on research studying Shewanella pathogenesis. Thus, I encourage you to revise your manuscript, accordingly, taking into account all of the concerns raised by both reviewers.

While the concerns of the reviewers are relatively minor, this is a major revision to ensure that the original reviewers have a chance to evaluate your responses to their concerns. There are not too many suggestions; thus, it should not take much effort to address these concerns to greatly improve your manuscript.

I look forward to seeing your revision, and thanks again for submitting your work to PeerJ.

Good luck with your revision,

-joe

·

Basic reporting

The present manuscript "Pathogenic strains Shewanella putrefaciens contain plasmids that are absent in the probiotic strain Pdp11" is a really interesting work that investigate the relation between plasmids and pathogenic/probiotic bacteria character.
Nowadays, the use of probiotic is being essential as prophylactic strategies against infectious diseases in aquaculture, but the essential characteristic at genomic level that confers probiotic feature to a microorganisms is still unknown. Unraveling these mechanisms in the Shewanella putrefaciens strain Pdp11, a very well-known probiotic, will help for further investigations in other probiotics.

Experimental design

The experimental design is well defined and the methods are described with sufficient detail and information to replicate. The used techniques and methodologies are completely suitable to answer the planned questions.

Validity of the findings

The shown results support the conclusions.

Additional comments

The investigation is really worthy and the manuscript is suitable for publication.

Reviewer 2 ·

Basic reporting

The manuscript is well written and organized.

However, a critical point is that the manuscript is not complying with the requirements of PeerJ regarding the availability of raw data and metadata. Raw data of plasmids sequencing information should be made available. According to PeerJ rules, authors can make raw data available by 2 ways:
1. Upload to PeerJ as a supplemental file.
2. Upload to an online repository and submit access details (required for files over 30MB).

Experimental design

The manuscript presents a study on the isolation, sequencing and genomic analysis of two plasmids occurring in two bacterial strains of pathogenic Shewanella putrefaciens. It fits the aims and scope of PeerJ, and is of interest for the scientific community working on fish pathogens.

The methodologies used are adequate and well described, allowing replication studies, with exception of a few clarifications that authors should do, to improve the overall quality of the manuscript before acceptance for publication.

Validity of the findings

The authors should further elaborate on their discussion of the fact that plasmids were only detected in 2 out of five pathogenic strains (Lines 319-320): does this means that plasmids might not be carriers of virulence factors in S. putrefaciens (being probably chromosomally encoded) or is it a technical matter because the authors used only one extraction methodology (commercial kit)?

Considering that commercial kits for plasmid extraction are optimized for artificial plasmids (cloning vectors), usually purified from E. coli, have the authors tried other extraction methods?

Authors must clearly state that the 2 plasmids detected might be responsible for the pathogenicity of strains SH4 and SH12 but the absence of such plasmids is not guaranteeing probiosis, as other pathogenic strains did not contain any plasmids. Without such clarification the sentence in Lines 319-320 “…the absence of plasmids in S. putrefaciens strain Pdp11 could underlie its probiotic nature, but not of strains SH4 and SH12, which presented one plasmid each, pSH4 and pSH12, respectively…” might be misinterpreted by the readers.

Regarding the sentence in Line 377: if using genetic engineering would the GMO probiotic still be valuable in EU? Do the authors believe this is the future, improving probiosis by genetically modifying available strains? The authors could add something on this to improve their discussion.

Additional comments

Lines 64 and Line 551: check spelling as some letters appear as capital lettering in : SÁenz De RodrigÁÑez et al. 2009;

Line 107: the word agitation should be removed as the authors are describing growth on agar plates in that sentence.

Line 111: I believe that the authors mean that plasmids can be integrated in the bacterial chromosome, not in the bacterial genome, as if present, both integrated plasmids (episome) and free circular plasmids are part of the genome. Thus genome here should be corrected to chromosome. Also, note that the majority of plasmids are extra-chromossomal elements; natural integration is a rather rare phenomenon. Integrative plasmids used in microbial genetics studies are usually suicidal (not able to replicate) - the review Plasmid persistence: costs, benefits, and the plasmid paradox, by Carrol & Wong is worth reading. https://doi.org/10.1139/cjm-2017-0609

Lines 316, 352, 365: for mobile genetic element (MEG) should be (MGE)

Figure 2 – although the authors have in the legend: “…The sizes of the individual plasmids contained in strains SH4 and SH12 as well as the seven plasmids of the positive control E. coli O157, grown in LB broth, are indicated...”the figure only shows the sizes of the Molecular Weight Marker, not the sizes of the plasmid bands. The E.coli plasmids bands sizes should be added to the figure, since only 5 bands are clearly visible, the presence of the sizes will help readers on correct figure interpretation.

Supplemental Figures S1 and S2 have poor legibility: higher resolution figures that allow full reading should be provided

---

## Round 0.2 · accepted · Accept

Dear Dr. Maqueda and colleagues:

Thanks for revising your manuscript based on the concerns raised by the reviewers. I now believe that your manuscript is suitable for publication. Congratulations! I look forward to seeing this work in print, and I anticipate it being an important resource for researchers studying Shewanella pathogenesis. Thanks again for choosing PeerJ to publish such important work.

Best,

-joe